# Neglected Diseases—Parasitic Infections among Slovakian Children from Different Populations and Genotypes of *Giardia duodenalis*

**DOI:** 10.3390/microorganisms10020381

**Published:** 2022-02-06

**Authors:** Júlia Šmigová, Viliam Šnábel, Serena Cavallero, Ľubomír Šmiga, Jindřich Šoltys, Ján Papaj, Ingrid Papajová

**Affiliations:** 1Institute of Parasitology, Slovak Academy of Sciences, Hlinkova 3, 040 01 Košice, Slovakia; bystrianska@saske.sk (J.Š.); snabel@saske.sk (V.Š.); soltys@saske.sk (J.Š.); 2Department of Public Health and Infectious Diseases, Sapienza University of Rome, Piazzale Aldo Moro 5, 00185 Rome, Italy; serena.cavallero@uniroma1.it; 3Department of Breeding and Diseases of Game, Fish and Bees, Ecology and Cynology, University of Veterinary Medicine and Pharmacy in Košice, Komenského 73, 041 81 Košice, Slovakia; lubomir.smiga@uvlf.sk; 4Faculty of Electrical Engineering and Informatics, Technical University in Košice, Letná 1/9, 042 00 Košice, Slovakia; jan.papaj@tuke.sk

**Keywords:** neglected diseases, helminthosis, protozoosis, children, *Giardia duodenalis* genotypes

## Abstract

Children are most prone to parasitic infections. The objectives of the study were to examine the occurrence of parasitic infections in children from different populations and to perform molecular characterization of human *Giardia duodenalis* isolates. We examined 631 stool samples from Roma and non-Roma children for the presence of parasitic developmental stages. Samples were collected from three eastern Slovakia districts. The ages of the children ranged from 1 months to 17 years. Subsequently, the molecular characterization of human *G. duodenalis* isolates by PCR detected triosephosphate isomerase (*tpi*) and beta-giardin (*bg*) genes was performed. The overall prevalence of parasitic infection was 19.8%. *Ascaris lumbricoides* eggs were the most frequent, with an occurrence of about 13.8%. *G. duodenalis* cysts were present in 6.3% of samples. *G. duodenalis* isolates obtained from 13 children were subjected to DNA sequencing with *tpi* and *bg* genes. Five isolates were categorized as bearing subassemblage BIII, the three isolates as subassemblage BIV, one person was infected with a mixture of subassemblages BIII and BIV, four children had subassemblage AII, and one isolate revealed a structure corresponding with subassemblage AI. Our work is proof that poverty and poor hygiene contribute the most to public health problems associated with neglected parasitic diseases.

## 1. Introduction

The WHO (World Health Organization) recently reported that zoonotic parasitic diseases are increasingly influencing human populations due to the effects of globalization, climate change, reduction in animal species and habitat variability [1]. The biggest polluters of the environment are people and their activities. The occurrence of various parasitic species is mainly influenced by climatic conditions and the survival ability of parasitic eggs and (oo)cysts in the outdoor environment.

The growing number of people is exacerbating social and economic disparities, leading to an increase in the number of people living on the edge of poverty, lower living standards and segregation, and to deteriorating health and a higher frequency of parasitic infections, especially in areas with the poor hygiene or in areas with insufficient infrastructure. In Slovakia, these places largely represent marginalized settlements on the outskirts of towns and villages, where most of these inhabitants do not have access to basic sanitation facilities or drinking water.

According to the Atlas of Roma communities elaborated by Mušinka et al. [2], 168,940 Roma people live in 803 marginalized or segregated settlements, which represents around 38% of the Roma population of Slovakia. As much as 30.5% of all segregated communities lack access to a public water supply system, 23.7% of dwellings use their own wells as the sources of water, and 15% use other sources of drinking water (natural springs, streams, etc.) [2]. Public sewerage is least accessible; only 41.7% of the settlements have access to the public sewer system. Gas is unavailable in 16.5% of settlements, and water pipes in 17.4% of settlements [2,3]. Poor socio-economic conditions, an unhealthy lifestyle, and barriers precluding access to healthcare are factors that influence the Roma population in settlements and lead to an increased prevalence of metabolic syndrome and its components: kidney disease, viral hepatitis B and E, tuberculosis, and some parasitic diseases [4].

In the Roma settlements of Slovakia, parasitic infections in children and dogs and their connection with soil contamination due to various hygiene standards have recently been analysed [5]. In this survey, infections in children and dogs occurred in all examined localities, while a lower prevalence of parasitic developmental stages in the soil was found in a village with a higher socio-economic standard of living, better personal and communal hygiene, and better care of dogs.

Children are the most susceptible group because parasitic diseases are associated with poor hygiene habits and an underdeveloped immune system in this age group [6]. The health disorders of gastrointestinal parasitosis (protozoa and metazoa) are often malabsorption, malnutrition, chronic anaemia, cognitive impairment and inability to prosper [7]. The most common parasitic infections are infections caused by soil-transmitted helminths *Ascaris lumbricoides*, *Trichuris trichiura*, *Ancylostoma duodenale* and *Necator americanus*, and infections caused by water-borne protozoan parasites such as *Giardia duodenalis* and *Cryptosporidium* spp.

*G. duodenalis* was first discovered by Antonie van Leeuwenhoek more than 300 years ago. In 2006, the WHO recognized giardiasis as a neglected disease associated with poverty and worsening development and socio-economic improvement [8]. In developing countries, it is estimated that approximately 200 million people suffer from symptomatic giardiasis [1]. The protozoan *G. duodenalis* is currently considered to be a multi-species complex divided into eight (A to H) genetically distinct assemblages, which differ in their specificity for hosts [9]. Assemblages A and B are associated with human infections and are considered zoonotic [9]. Furthermore, molecular studies have classified sub-assemblages in assemblages A and B. Subassemblage AI is being recorded in both animals and humans and is mainly of zoonotic transmission, while subassemblage AII is predominantly typical for humans with anthroponotic transmission, and subassemblage AIII is characteristic for animals only. Subassemblages BIII and BIV are regularly reported in humans, companion animals, and wild animals. Assemblages C and D were recorded in dogs and other canines, E in hoofed livestock, F in cats, G in rats, and H in marine mammals [10]. Human infections caused by *G. duodenalis* are not included in the list of 20 tropical diseases that are a priority of the WHO. Although these infections are generally considered less of a priority in regard to the protection of public health, and, their increasing prevalence worldwide has demonstrated significant impact on morbidity and even mortality [11].

In Slovakia, 44 diseases are monitored in the National Monitoring Report “Summary Report on Zoonoses, Food and Water-borne Infections in the Slovak Republic”. However, official data on human intestinal parasites still do not exist in our country. Last year, 17,067 cases of human disease caused by viruses, bacteria, parasites and prions were reported in the Slovak Republic, as stated in the “Report on Zoonoses, Alimentary Diseases and Waterborne Diseases in the Slovak Republic in 2020” [12]. This is a significant decrease compared to 2019, when 26,999 cases of human infections were recorded. The most common epidemics in 2019 and 2020 were due to salmonellosis, campylobacteriosis and rotaviruses. Official data on the prevalence of intestinal parasites in Slovakia are not available.

The initial aim of this study was therefore to examine stool samples from a group of Slovak children without any clinical signs for the presence of gastrointestinal parasites and to collect data on possible associated risk factors. We have surveyed the prevalence of parasitic infections in children coming from distinct populations in eastern Slovakia. Subsequently, the molecular characterization of human *G. duodenalis* isolates was conducted by using DNA sequencing of triosephosphate isomerase (*tpi*) and beta-giardin (*bg*) genes, which are long established markers to distinguish between all *G. duodenalis* types and subgroups.

## 2. Materials and Methods

The study population consisted of children and adolescents from various towns and villages in eastern and southern Slovakia and segregated Roma settlements in eastern Slovakia. A total of 631 stool samples were collected from children and adolescents aged from 1 month to 17 years. The average age of the examined children was 6.8 years.

### 2.1. Collection of Samples

After parents or legal guardians have signed the informed consent forms, the stool samples were collected in plastic containers. Each of the 631 stool sample containers was identified by a specific identification number and supplemented with age, gender, residence, and compliance with the majority or minority population. Of the children surveyed, 349 belonged to the majority population and 282 to the minority population. A total of 311 boys and 320 girls were examined. The children were divided into three groups according to age: the first group consisted of preschool children from 1 month to 5.9 years, the second group of compulsory school children from 6 years to 14.9 and the third group from adolescents from 15 years to 17.9 (Table 1).

The survey was conducted in collaboration with parents, guardians and social workers responsible for segregated settlements. Social workers helped researchers obtain samples from children in segregated settlements. Parents in the majority population were asked to take a stool sample from the children for a preventive check-up. The children in this study were clinically healthy, without any clinical signs of parasitic infection (diarrhoea, nausea, etc.). Parents and/or guardians were asked to sign a written informed consent and participation of everyone individual was voluntary. Individuals who agreed to participate were given an empty plastic container with a unique identification number and instructed on how to take a small sample (~10–15 g) the next morning.

### 2.2. Coprological Diagnostic Methods

All samples were stored without any preservation at 4 °C and were transported directly to the laboratory of the Institute of Parasitology of the Slovak Academy of Sciences in Košice for further parasitological examination. The entire analysis was performed within 24–48 h. The stool sample volume was (~10–15 g) from each individual. Stool samples were examined macroscopically for the detection of proglottids and then microscopically screened for the presence of worm eggs and protozoal (oo)cysts. Each sample was examined microscopically using two methods: by a commercially available kit (Paraprep L; Mondial, France) and by the sodium acetate, acetic acid and formalin (SAF) concentration technique.

### 2.3. Diagnostic Techniques

Paraprep L (Mondial, France) is a commercially available kit. Briefly, using this kit, for 0.5 g of each stool sample, 2 mL of ethyl acetate solution was added to 6 mL of 10% formalin in a mixing chamber. Subsequently, the tube was connected to a conical collection tube with the filter in between. The mixed content was incubated for 24 h at room temperature. After 24 h of incubation at room temperature, the tube was centrifuged at 1000 rpm for 1 min, the supernatant was discarded and the sediment examined microscopically with a Leica DM 5000B light microscope (Leica Microsystems; Wetzlar, Germany) at 100× and 400× magnification.

The SAF concentration technique is often used in European reference laboratories for the detection of eggs of gastrointestinal parasites. Briefly, using this technique, ~1 g of stool sample was used for examination. The SAF fixed stool samples were resuspended and strained through a medical gauze into a centrifuge tube. After centrifugation at 2500 rpm for 3 min, the supernatant was discarded, and the sediment resuspended in 7 mL of 0.85% NaCl. After the addition 2–3 mL of diethyl ether, the tube was closed with a rubber stopper, shaken vigorously for ~30 s and then centrifuged at 2500 rpm for 3 min. After the second centrifugation, the three top layers were discarded, and the resulting sediment was examined microscopically with a Leica DM 5000B light microscope (Leica Microsystems; Wetzlar, Germany) at 100× and 400× magnification for the presence of helminths and intestinal protozoa.

### 2.4. Molecular Diagnostic Methods

#### 2.4.1. Extraction of DNA

DNA was extracted directly from fresh stool samples. A 200 mg of stool material aliquot sample was homogenized with a Qiagen TissueLyser (Qiagen, Hilden, Germany). Then, DNA was purified by Quick-DNA™ Faecal/Soil Microbe Miniprep Kit (Zymoresearch, Irvine, CA, USA) according to the manufacturer’s instructions and stored at −20 °C.

#### 2.4.2. Amplification of DNA Nested PCR

Amplification of the triosephosphate isomerase (*tpi*) gene and beta-giardin (*bg*) gene nuclear targets, commonly used for *Giardia* genotyping studies, was performed by a nested PCR method. The resulting fragments were of 477 bp and 485 bp in length, respectively.

For the nested PCR to amplify the *tpi* fragment, the amplification reaction of 25 μL total volume consisted of 1X CoralLoad PCR Buffer, 1 mM MgCl_2_, 200 μM of each dNTP, 500 nM of each of external primers AL3543 [5′-AAATIATGCCTGCTCGTCG-3′] and AL3546 [5′-CAAACCTTITCCGCAAACC-3′], 2.5 U of HotStarTaq Plus DNA Polymerase (Qiagen, Germany) and 5 μL of DNA for the primary amplification. PCR was performed in 35 cycles of 94 °C for 45 s, 50 °C for 45 s and 72 °C for 60 s, with an initial hot start of 94 °C for 5 min and final extension of 70 °C for 10 min, and 4 °C holds in MyCycler™ Thermal Cycler System (Bio-Rad Laboratories, Berkeley, CA, USA). For the second PCR reaction, 1 μL of PCR product from primary reaction and pair of internal primers AL3544 [5′-CCCTTCATCGGIGGTAACTT-3′] and AL3545 [5′-GTGGCCACCACICCCGTGCC-3′], were added. PCR conditions for the second reaction were identical [13].

Amplification of the *bg* fragment by the nested PCR with was performed in 25 μL volume containing 1X CoralLoad PCR Buffer, 1 mM MgCl_2_, 200 mM of each dNTP, 500 nM of each of primers B3: GAA CGA ACG AGA TCG AGG TCC G and B4: CTC GAC GAG CTT CGT GTT [14], 2.5 U HotStarTaqPlus DNA Polymerase (Qiagen, Germany) and 5 μL of DNA for the primary amplification. The PCR reaction was carried in a MyCyclerTM Thermal Cycler System (Bio-Rad Laboratories, Berkeley, CA, USA). It consisted of 1 cycle of 96 °C for 5 min, followed by 55 cycles of 94 °C for 30 s, 65 °C for 30 s and 72 °C for 60 s, with the final extension of 72 °C for 7 min and 4 °C holds in the same thermal cycler. A reference *G. duodenalis* isolate was used as a positive control and water was used as a negative control. Amplicons were separated on 1.5% agarose gel stained by GelRed^®^ Nucleic Acid Gel Stain (BIOTIUM, Fremont, CA, USA) and TAE buffer (40 mM Tris, pH 7.8, 20 mM acetic acid, 2 mM EDTA).

#### 2.4.3. Sequencing of DNA

For DNA sequencing, the amplification products obtained from the PCR reactions were purified with the ISOLATE II PCR and Gel Kit (Bioline, London, UK) and sequenced in an Avant 3100 sequencer (Applied Biosystem, Waltham, MA, USA). Both strands used the same set of internal primers as in respective PCR assays. DNA sequencing was performed by a sequencing facility at the University of Veterinary Medicine, Košice, Slovakia. The obtained DNA sequences in the partial *bg* and *tpi* genes were compared with the reference sequences of *G. duodenalis* in the GenBank database by the nucleotide BLASTn program [15]. Sequences were sorted by similarity and aligned using the Clustal Omega tool [16]. Phylograms were constructed by the MEGA7 software [17] using the maximum-likelihood (ML) method with 1000 bootstrap pseudoreplicates. A TN93 substitution model with gamma-distributed rate heterogeneity [18] was selected based on the maximum likelihood evaluation of possible substitution models using the Modeltest in MEGA.

The gene sequences for 13 isolates examined in *tpi* were deposited in GenBank^®^ under accession numbers OL456145-OL456151 and OL474006-OL474011, and under accession number OL581604 for isolate examined in *bg*.

### 2.5. Statistical Analysis

Statistical analysis of the obtained results was performed using the statistical product StatSoft, Inc. (2007). STATISTICA (software system for data analysis), version 8.0. and StatSoft Statistica 12 and Microsoft Excel (Office 365). The potential risk was calculated using the exchange rate (OR) as a relative chance and the relative risk using the risk rate (RR). Crosstables and Chi-square tests were used to calculate correlations between sex, the children’s residence belonging to a majority or minority population, and age-class. Differences were considered statistically significant at *p* < 0.05 and at *p* < 0.001.

## 3. Results

A total of 631 stool samples from minority and majority child populations were examined for the presence of parasite eggs. The children’s age ranged from 1 months to 17 years. The overall prevalence of infection was 19.8% (125/631). The most frequently detected parasites were roundworm eggs of *A. lumbricoides* in 13.8% and cysts of *G. duodenalis* in 6.3% of samples. Eggs of *Trichuris trichiura*, *Hymenolepis nana*, and *Enterobius vermicularis* were also present in the children’s stool, and their occurrence was confirmed in less than 2% of the samples. *Blastocystis hominis* cysts were present in one stool sample (Table 2).

The most infected group of the examined children were preschool children aged 1 month to 5.9 years. Preschool children were infected with developmental stages of endoparasites at a frequency of 21.6% (69/319). Compulsory school children aged 6 to 14.9 years, 20.3% (54/266) and adolescents over 15 years of age were infected in only two cases (4.3%, 2/46).

Of the children examined, 349 belonged to the majority population, and 282 to the minority population. A total of 311 boys and 320 girls were examined. The presence of parasitic infections was detected in 10 (2.87%) children from the majority group and 115 children from the minority population (40.8%). Six positively tested children lived in rural areas from the majority population, and four positive children came from urban areas. The children from the majority group were infected with four different parasite species, in which *A. lumbricoides* eggs were most frequently detected (7 cases). One positive case of *T. trichiura*, *E. vermicularis,* and *G. duodenalis* was also confirmed. Only single infections with one parasitic species were found in the sample set surveyed.

The children in the majority group were infected with four different species of parasites, in which *A. lumbricoides* eggs were most frequently detected (7 cases).

In the minority population, 86 children came from segregated settlements, 112 from urban areas, and 84 from rural areas. 354 children were examined from urban areas, and parasitic infection was detected in 37 of them (10.5%).

A total of six different parasite taxonomic units, such as *A. lumbricoides, B. hominis, E. vermicularis, G. duodenalis, H. nana* and *T. trichiura*, have been identified in stool samples from the minority children’s population.

The most frequently detected were eggs of *A. lumbricoides* (28.4%) and cysts of *G. duodenalis* (13.8%). *T. trichiura* eggs were identified in 2.5% of samples, and the prevalence of *H. nana* tapeworm eggs was 2.1%. *B. hominis* cysts were present in one sample. Furthermore, *E. vermicularis* eggs were diagnosed in 1% of the samples. In contrast to children from the majority population, mixed infections were found in children from the minority population in up to 17 cases. A total of 15 individuals were co-infected with two different intestinal parasites. In 12 cases, *A. lumbricoides* with *G. duodenalis* cyst was detected in stool samples. In two cases, cysts of *G. duodenalis* and eggs of *H. nana* were detected, and in one case, the embryos of *A. lumbricoides* and *E. vermicularis* were found. Co-infection with three eggs of *A. lumbricoides* + *G. duodenalis* + *E. vermicularis* gastrointestinal parasites was detected in one sample. In addition, co-infection with four different developmental stages of *A. lumbricoides* + *G. duodenalis* + *T. trichiura* + *H. nana* was recorded in one sample.

Minority children from urban areas were most often infected with *A. lumbricoides* eggs (27 cases observed). In nine cases these children were infected with *G. duodenalis* cysts and in one case with *T. trichiura*, *E. vermicularis* and *H. nana.*

Children from rural areas were affected by a parasitic infection in 24.1% of cases. This corresponds to 46 positive children out of 191 samples examined. The most common parasitic eggs found in stool samples of these children were *A. lumbricoides* eggs, detected in 31 cases. *G. duodenalis* cysts were detected in 14 stool samples. *E. vermicularis* eggs were found in two cases in children’s stool samples from rural areas from the minority population, whereas *T. trichiura* eggs and *B. hominis* cysts were presented in one sample.

The most serious situation in terms of parasitic infections was found in children from segregated settlements. Of the 86 children examined, up to 42 were positive, which represents a prevalence of 48.9%. *A. lumbricoides* eggs were most commonly found in children from segregated settlements, as well as in minority children from urban and rural areas (in 22 cases). *G. duodenalis* cysts were found in 16 cases, the most frequently of all minority groups. *T. trichiura* eggs were found in five cases of samples from segregated settlements and *H. nana* eggs were also found in five cases.

Differences between the prevalence of parasitic infection in children from majority and minority populations are statistically significant (Table 3). Children from the minority population were significantly more often infected with parasites than children from the majority group (X^2^ 141.14; *p* < 0.00001). The probability of infection (OR 23.344) with parasitic developmental stages was 23.3 times higher in the minority population than in the majority group. The development of parasitic infection was therefore identified as a risk factor for the minority population (RR = 14.32).

Differences between the prevalence of parasitic infections in children from urban, rural areas and areas of marginalized groups were statistically significant (X^2^ − 67.33; *p* < 0.00001; Table 3). The probability of infection was up to 8.17 times higher (OR 8.178) in children from marginalized groups than in children from urban areas. Compared with infections in children from marginalized settlements and rural areas, marginalized children were infected three times more often than children in rural areas (OR 3.009).

Differences between the prevalence of parasitic infection in children from urban and rural populations were also statistically significant (X^2^ = 16.82; *p* < 0.01) (Table 3). The probability of infection was 2.7 times higher in children from rural areas than in children from urban areas (OR 2.718). At the same time, the rural environment was evaluated as a risk factor for the development of parasitic infection (RR 2.304).

### Molecular Diagnosis of G. duodenalis in Children

Forty-four sample cysts were detected by coprological examination. Since the size of the 20 samples was insufficient, the parasitic DNA was isolated from 24 stool samples. Two different sections of the triosephosphate isomerase (*tpi*) and β-giardin (*bg*) genes were used to amplify the DNA. Amplification of parasitic DNA was successful in 24 isolates. Amplification in the *tpi* gene regions was successful in 24 isolates of protozoan DNA. In addition, only two DNA isolates were sufficiently amplified in the ß-giardin region.

Thirteen isolates from children from three districts in eastern Slovakia were finally examined by DNA sequencing of the *tpi* and *bg* partial genes. In nine isolates (69.2%), assemblage B was detected, within which five isolates profiles corresponded to subassemblage BIII, three isolates profiles checked with subassemblage BIV, and one isolate showed a discordant genetic structure revealing subassemblages BIV in *tpi* and BIII in *bg*. In four isolates (30.8%), assemblage A was recorded; isolates of three profiles clustered with subassemblage AII, and one isolate was categorized as having a structure of subassemblage AI.

For the *tpi* partial gene (477 bp), the samples clustered in subassemblages BIII and BIV were distinguished by good bootstrap support (89%). The resulting ML phylogram is shown in Figure 1. In a sample set typed as BIV subassemblage, three isolates, JS-Lu268 and JS-Lu were obtained from humans in the Košice district (locality of Luník IX) and JS-Rv141 from the Rožňava district (both located in southeastern Slovakia). They had a pattern characteristic of the reference BIV sequence (GenBank entry AF069560), which differed from previously sequenced BIV isolates (195HuRT, MG515175, and 17HuRP, MG515174) from Slovak children [19] in two nucleotides.

Sample JS-Lu233 showed a unique structure differing from BIII in four sites (55C/T, 132C/T, 174G/A, 393G/A) and from BIV in one site (132T/C). One nucleotide position (132Y) showed double peaks composed of C/T in the sequence chromatograms (see Table 4 describing the *tpi* polymorphism). Overall, JS-Lu233 grouped with samples attributable to the BIV subassemblage but formed a separate branch in the resulting phylogram (Figure 1).

In a sample set typed as BIII subassemblage, two examined isolates matched the reference BIII structure, namely JS-Lu268 and JS-Lu269 originating from the Košice district. The other three isolates (JS-Rv54, JS-Lu146, JS-Pe61) had between one to five single nucleotide polymorphisms (SNP´s). All three isolates exhibiting this BIV polymorphism shared a nucleotide substitution 126G/A, according to a BLAST search commonly reporting for *G. intestinalis* worldwide. A representative isolate in the phylogram carrying this variation is described by Ankarklev et al. [20] and originated from Sweden (JN579668).

Besides this polymorphism, isolate JS-Lu146 (Košice district) also manifested double C/T peaks (resulting in a Y signal) at diagnostic BIII/BIV sites. In the JS-Rv54 isolate (Rožňava district), substitutions 28A/G, 55C/T, 150A/G were detected, which were accompanied by a double R signal (composed of A/G nucleotides) at position 159. The closest genetic pattern to JS-Rv54 in GenBank was found in five *G. intestinalis* isolates from Egypt (KR260581, AB781124), Brazil (KF922896), West Bank, Palestine (AB480874), and Turkey (MT166373), differing in three nucleotides. The latter isolate described by Sarzhanov et al. [21] was included in the presented phylogram.

Considerable intra-subassemblage polymorphism within BIII was also recorded in isolate JS-Pe61 from the Bardejov district (Petrová, northeastern Slovakia), which showed nucleotide exchanges 46A/C, 192 C/T, 222C/T, and double C/T signal at position 105 (Table 4). The first two polymorphisms are unique for *Giardia* parasites, whereas the 222C/T variation was previously detected in one human isolate from Spain (KX668290).

In comparison to five *tpi* profiles (assigned to as P1-P5) of subassemblage BIII, before reported in Slovak children [22], three new genotypes harbouring by humans (carrying by JS-Rv54, JS-Lu146, JS-Pe61 isolates) were recorded in the present study. Only the P3 profile (corresponding to the BIII reference structure), observed here in the two samples JS-Lu268 and JS-Lu269, has been earlier documented in Slovakia.

In four samples defined through *tpi* sequences as belonging to the assemblage A, isolates JS-Lu152, JS-Lu236 (both Košice district) and JS-Pl266 (Plešivec, Rožňava district) were assigned to subassemblage AII, and one isolate (Krásnohorské Podhradie, Rožňava district) to subassemblage AI. All A-typed samples were identical to the referenced sequences established for subtypes AI, AII.

No geographically induced preference towards any of the assemblages A, B was detected when comparing their distribution in two districts (Košice, Rožňava) with several analyzed samples in *tpi*.

In the β-giardin (*bg*) partial gene (485 bp), isolate JS-Lu233 showed a profile with four nucleotide substitutions (90G/A, 216T/C, 426C/T, 471T/C) characteristic of the BIII reference structure and only two substitutions typical of the BIV structure (135G/A, 219C/T). In this gene, JS-Lu233 (see ML phylogram for *bg* in Figure 2) overall clustered with a BIII-like structure (unlike the BIV affiliation in *tpi*). These discordant subtype results might indicate mixed infection with different *G. duodenalis* genotypes or the occurrence of genetic recombination leading to allelic sequence heterozygosity (ASH, see Discussion for further explanation). Although partially representing the intermediate structure of BIII/BIV, this pattern has been relatively commonly recorded in various countries on four continents according to BLAST search (Egypt, Kenya, Ethiopia, Iran, Bangladesh, Brazil, and Norway). The presented *bg* phylogram included the Norwegian isolate BG-Ber7 examined by Robertson et al. [23] from this sequence type (Figure 2). In *bg*, two different BIII profiles, attributable to samples MAR3-SK (MZ160204) and EST3-SK (MZ160200) from Medzev (Košice district), were previously documented in clinical samples from Slovakia. Genetic characterization of further *Giardia* isolates from Slovak children in the *bg* gene was impossible due to unsatisfactory sequencing results.

## 4. Discussion

In the present study, we have primarily examined the circulation of parasitic infections in the human population focused on the most vulnerable group of people: children. Of the 631 stool samples analysed, 125 children were positive, representing a prevalence of 19.8%, which is comparable to the analysis of Solovič et al. [24] carried out on a marginalized Roma community in eastern Slovakia in 2011. Similar results were also reported for children from the marginalized population in eastern Slovakia in more recent studies of Rudohradská et al. [25] and Pipiková et al. [5]. This indicates that the epidemiological situation regarding parasitic infections has not improved significantly in Roma settlements of Slovakia over the last decade, despite a number of state intervention programs to increase hygiene coverage and infrastructure in marginalized communities. Safi et al. [26] reached similar findings in a survey on stool samples from Afghan compulsory school children, which revealed the presence of parasitic developmental stages in a 26.6% prevalence. A much higher prevalence was found, e.g., in Malaysia, where up to 56.3% of examined children under the age of 18 were positive for parasitic infections [27].

We detected six different taxonomic units of parasites in the examined stool samples in the current report. The most common parasitic eggs in the samples were *A. lumbricoides* in 13.8%, which corresponds to the results of Safi et al. [26] in Afghanistan and Hassan et al. in Malaysia [27]. Protozoal cysts *G. duodenalis*, which are the most common cause of diarrheal disease in some countries [28], were detected microscopically in 6.3% of stool samples we examined.

Contrary to our results, *G. duodenalis* infections were observed only in a prevalence of 1.3% in Greece [29]. The eggs of *T. trichiura* were also found in our samples (1.26% prevalence), which corresponds to the findings from the Philippines [30], where these parasitic infections were detected in 1.67% of children. A slightly higher prevalence of *T. trichiura* eggs (1.8%) has been reported from Denmark in children adopted from developing countries [31]. An interesting finding in our study was four cases of infection with *E. vermicularis*, which is, e.g., in Thailand one of the most common parasites, where the prevalence is around 5.8% [32]. However, the method of perianal adhesions was not used in our work. Nevertheless, these parasitic eggs were present in almost 1% of samples, which indicates a high prevalence of this infection in the examined children. The eggs of tapeworm *H. nana* were found in stool samples obtained from 1% of children. The results of our study suggest that tapeworm infection in Slovak children is much lower compared to children from most of other countries. *H. nana* was detected in 4.4% of adopted children in Denmark [31]; in Thailand, they reach *H. nana* exposure up to 13.1% [33]. In Peru, eggs of *H. nana* were detected up to 17.4% [34]. We assume that they were *H. nana* eggs, given that *H. diminuta* infection would require the consumption of cysticercoids from insects, whereas infection by *H. nana* tapeworm may be caused without intermediate host. *H. nana* eggs unlike *H. diminuta* eggs, the embryophore has no conspicuous knobs and filaments at the poles [35]. In our study in children from urban areas, parasitic infections were reported in 10.45% of cases, while in rural areas the prevalence of the parasite was more than doubled (24.1%). The same results have been reported from urban and rural areas in Poland, where Bitkowska et al. [36] detected 19% prevalence of developmental parasitic stages in stool samples from children in rural areas and 10.4% in urban areas.

Parasitic infections were more frequently diagnosed in children from marginalized Roma settlements. Of the 86 children, up to 48.8% were positive for the presence of parasitic developmental stages. This may be due to lower levels of personal hygiene, socio-economic disadvantage, accumulation of biowaste and closed contact with feral populations of cats, dogs, rodents and insects. In Europe, a comparatively high prevalence of parasitic infection was also diagnosed by Gualdieri et al. [37], who studied immigrants in southern Italy living in segregated settlements. Parasitic infections were detected in 61.9% of samples. Our study is proof that parasitic infections pose a serious problem to human health, especially in children from a minority Roma population. In these children, parasitic infections were detected in 115 of 282 children examined, which represents a prevalence of up to 40.8%. Our results are comparable with the results of parasitic examinations in children from developing countries, who were adopted in Europe and parasitic infections were present in up to 54% of them [31]. The high prevalence of parasitic infection recorded in Roma settlements in our study is comparable to the results from developing countries. In Nicaragua, the prevalence of parasitic infections in children was up to 54.3% [38]. A higher prevalence of parasitic infections was reported in Malaysia, where parasitic infections were found in 56.3% of examined children [27]. The most common finding in 28.4% of Roma children was the roundworm eggs, which corresponds to Rudohradská et al. [25], who diagnosed roundworm eggs in 24.7% of Roma children. Similarly, Hassan et al. [27] diagnosed *A. lumbricoides* in 24.3% of children in Malaysia. We confirmed a high 13.8% prevalence of *G. duodenalis* cysts. Our findings are comparable to those of Ögren et al. [28], who found a 14% prevalence of *G. duodenalis*. In Italy, Gualdieri et al. [37] reported *G. duodenalis* cysts in 4.5% of examined immigrants.

The second most common helminth in Roma children observed in our work, *T. trichiura*, was present in 2.5% of all examined samples. This nematode is one of the most common parasites, e.g., in Thailand, where the prevalence is around 10.7% [39]. Comparable results were published from Afghanistan, where trichurid eggs were detected in 1% of the children examined. Tapeworm eggs of *H. nana* were found in 2.1% of samples from the minority children population. These results are comparable to those of Mohamed et al. [40], who diagnosed these eggs in 3.2% of children in Burkina Faso or in children from developing countries where hymenolepidosis occurred with a prevalence of 4.4% [31].

Based on our statistical results, we can state that children from the minority population were infected with parasites significantly more often than children from the majority population. Being a member of a marginalized population in eastern Slovakia is, thus, a risk factor for acquiring a parasitic infection. This is due to differences in socio-economic status and hygiene standards between the majority and minority populations of children. Statistical analysis also showed that preschool children are more often infected with parasites. In this age group, this is due to an underdeveloped immune system, poor hygiene habits, and more frequent contact with soil that may be infected by the developmental stages of parasites.

By employing the sequencing of *tpi* and *bg* genes, we classified nine isolates belonging to assemblage B and four isolates to assemblage A. In a global compilation study conducted by Sprong et al. [41], a relatively balanced distribution of BIII/BIV subassemblages in humans in Europe (49% BIII, 51% BIV) and Australia was found, unlike remaining continents. In the present study, we have also found a relatively proportional distribution in a set of nine isolates typed as B, with 61.1% of children infected with BIII and 38.9% with BIV (i.e., five BIII profiles, three BIV profiles, and one BIII/BIV profile). A balanced frequency of BIII (3 isolates) and BIV (4 isolates) in stool samples from children in eastern Slovakia was also confirmed by Pipiková et al. [19].

Detection of discordant genotypes when employing two or more genetic markers is a common phenomenon observed in *Giardia* parasites and is especially frequent in areas of high endemicity where the infection pressure and transmission intensity are high [42,43]. Globally, an infection with BIV in humans had occurred as often as an infection with a mixture of BIII/BIV (105 and 107 cases, respectively) [41]. According to the above report, mixing between subgroups BIII and BIV was found worldwide in 26.4% of isolates typed as B assemblage. In the present work, we found one (11.1%) human isolate (coded as JS-Lu233) displaying conflicting typing results (BIII using *bg*, BIV using *tpi*) within B assemblage. Given that a mixed template with a double peak at one of the differentiation sites (129C/T) for the BIII and BIV subgroups with *tpi* was detected, polymorphism in JS-Lu233 appears to be due to mixed infection with genetically distinct cysts rather than due to allelic sequence heterogeneity, i.e., sequence dissimilarity between different alleles of the same gene. The concerned isolate was obtained from a child in the Roma segregated settlement of Luník IX (Košice city), characterized by the low level of sanitation and environmental hygiene, where a higher occurrence of mixed infections due to high endemicity is expected.

For the A assemblage, the above report of Sprong et al. [41] confirmed that the three *G. duodenalis* subassemblages globally preferentially transmit within their primary hosts (AI in livestock, AII in humans, AIII in wildlife) and that these cycles do not interact substantially. Regarding Europe, Sprong et al. [41] assessed that the majority (75%) of human infections are linked to subassemblage AII and only 25% of infections are assigned to subassemblage A1 (sequences of 295 European human isolates were processed). A similar pattern was recorded in the present study. The subtype AII was identified in three children (accounting for 75% of A cases), predominating over the subtype AI detected in one child (25%). Accordingly, in the two previous molecular epidemiological studies on *Giardia* in humans from eastern Slovakia [19,22], all five samples were classified as assemblage A and were assigned to the AII subgroup, which is, in general, responsible for the maintenance of anthroponotic transmission [44].

Our work is proof that poverty and poor hygiene contribute most to public health problems. The risk of spreading endoparasites in various anthropogenically polluted ecosystems depends on the contamination of the environment with animal and human faeces. The cooperation of human and veterinarians, supported by the latest scientific knowledge and the cooperation of the general public, is essential to reduce this contamination and reduce the risk of infections, and, in particular, education and public awareness.

## Figures and Tables

**Figure 1 microorganisms-10-00381-f001:**
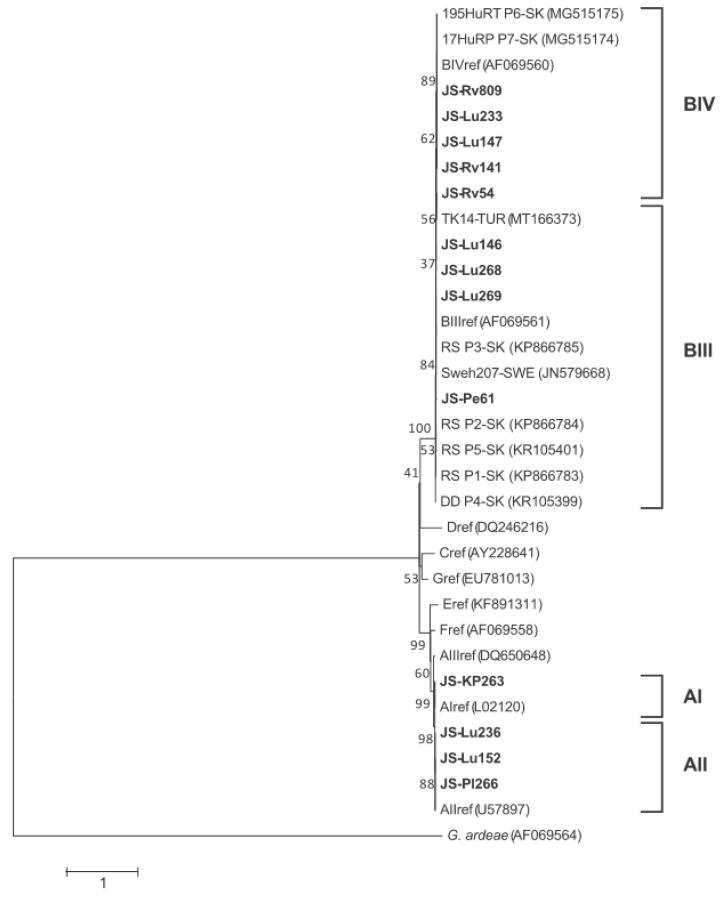
Maximum-likelihood phylogram generated from a partial *tpi* gene (477 bp) showing relationships between the examined Slovak isolates of *Giardia duodenalis* and GenBank-retrieved related sequences. The *G. ardeae* sequences from the grey heron (ref. AF069564) were used as an outgroup. The scale bar refers to a phylogenetic distance of 1.00 nucleotide substitutions per site. Numbers next to the branches indicate the bootstrap value calculated from 1000 pseudoreplicates. Geographical origins of the examined isolates in Slovakia: Lu Luník IX (Košice district, south-eastern Slovakia); Rv Rožnava city (southeastern Slovakia); Pl Plešivec (Rožňava district); KP Krásnohorské Podhradie (Rožňava district); Pe Petrová (Bardejov district, north-eastern Slovakia). Isolates containing P1-P7 in sample codes correspond to previously recorded profiles in Slovak humans for assemblage B using the *tpi* gene.

**Figure 2 microorganisms-10-00381-f002:**
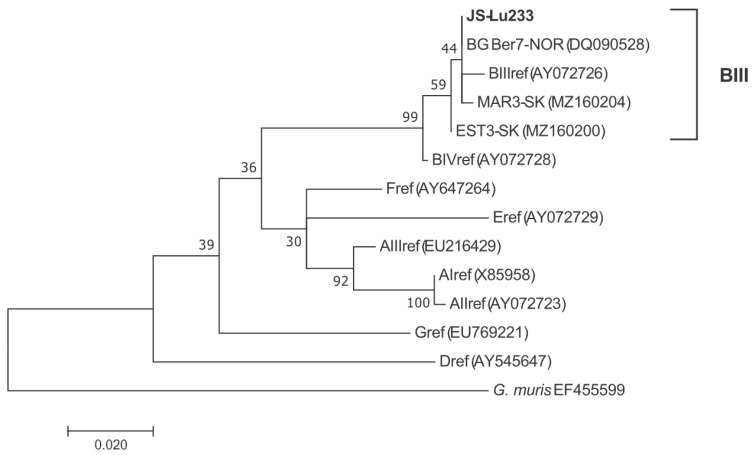
Maximum-likelihood phylogram generated from a partial *bg* gene (485 bp) showing relationships between the examined Slovak isolates of *Giardia duodenalis* and GenBank-retrieved related sequences. The *G. muris* sequences from the mouse (ref. No. EF455599) were used as an outgroup. The scale bar refers to a phylogenetic distance of 0.02 nucleotide substitutions per site. Numbers next to the branches indicate the bootstrap value calculated from 1000 pseudoreplicates. Geographical origin of the examined isolate in Slovakia: Lu Luník IX (Košice district, south-eastern Slovakia). Isolates MAR3-SK and EST3-SK correspond to previously recorded profiles in Slovak humans for assemblage A using the *bg* gene.

**Table 1 microorganisms-10-00381-t001:** Categories of origin, age classes, sex, and match with population.

Sample Origin	Majority						Minority					
	Girls			Boys			Girls			Boys		
	<5.9 *	6–14.9 *	15–17.9 *	<5.9 *	6–14.9 *	15–17.9 *	<5.9 *	6–14.9 *	15–17.9 *	<5.9	6–14.9 *	15–17.9 *
Urban	65	52	11	52	50	13	16	33	4	31	24	3
Rural	24	22	5	29	23	3	19	26	1	19	17	3
Settlement	0	0	0	0	0	0	31	8	3	33	11	0

* Age classes.

**Table 2 microorganisms-10-00381-t002:** The occurrence of parasitic developmental stages in children stool samples (*n* 631).

Detected Parasite Eggs	No. Infected	Prevalence (%)
*Ascaris lumbricoides*	87	13.8
*Trichuris trichiura*	8	1.3
*Giardia duodenalis*	40	6.3
*Enterobius vermicularis*	4	0.6
*Hymenolepis nana*	6	1.0
*Blastocystis hominis*	1	0.2
Total	125 *	19.8

No. infected positive samples, *n* the total number of samples, * includes 17 cases of mixed infection.

**Table 3 microorganisms-10-00381-t003:** Distribution of risk factors for positive stool parasitological examination.

			Positive (*n* = 125)		Negative (*n* = 506)			
			N	%	N	%	*X* ^2^	*p*
Sex	Male	311	70	22.51	241	77.49	2.81	0.09364
	Female	320	55	17.19	265	82.81		
Residence	Urban	354	37	10.45	317	89.55	Sg × U 67.33 *	0.00001
	Rural	191	46	24.08	145	75.92	R × U 16.82 *	0.00004
	Segregated settlements	86	42	48.84	44	51.16		
Population	Majority	349	10	2.87	339	97.13	141.14 *	0.00001
	Minority	282	115	40.78	167	59.22		
Age (years)	Preschool <5.9	319	69	21.6	250	78.4	P × A 7.66 *	0. 005634
	School Children 6–14.9	266	54	20.3	212	79.7	S × A 6.77 *	0.009233
	Adolescents 15–17.9	46	2	4.3	44	95.6		

* Statistically significant variables, *n* the total number of samples, N number of positive samples, *p p* value, Sg × U statistical comparison of prevalence of samples from segregated settlements and urban areas, R × U statistical comparison of prevalence of parasitic infections in rural and urban areas, P × A statistical comparison of prevalence of parasitic infections in preschool children and adolescents, S × A statistical comparison of prevalence of parasitic infections in school children and adolescents.

**Table 4 microorganisms-10-00381-t004:** Nucleotide polymorphism in B assemblages *of G. duodenalis* in the partial *tpi* gene.

Nucleotide Position within the *tpi* Gene													
Isolate	28	46	55	105	126	129	132	150	159	174	192	222	393
BIII ref. sequence	A	A	C	C	G	C	C	A	G	G	C	C	G
BIV ref. sequence	A	A	T	C	G	T	T	A	G	A	C	C	A
JS-Rv54 (*BIII subtype)*	** G **		** T **		** A **			** G **	** R **				
JS-Lu146 (*BIII subtype*)					** A **	** Y **	** Y **						
JS-Pe61 (*BIII subtype*)		** C **		** Y **	** A **						** T **	** T **	
JS-Lu233 (*BIV subtype*)			** T **			** Y **				** A **			** A **

Nucleotides highlighted in blue colour polymorphism in differentiating sites for BIII, BIV subassemblages. Nucleotides highlighted in red colour additional polymorphism in the other sites within subtypes.

## Data Availability

Data are available from the first author and the corresponding author upon reasonable request.

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
