# Peer review of "Neglected Diseases—Parasitic Infections among Slovakian Children from Different Populations and Genotypes of Giardia duodenalis"

_microorganisms, 2022, doi:10.3390/microorganisms10020381_

Round 1

Reviewer 1 Report

Review of the manuscript No. microorganisms-1558133 entitled "Neglected Diseases - Parasitic Infections among Slovakian 2 Children from Different Populations and Genotypes of Giardia 3 duodenalis"

The manuscript presents results of the epidemiological studies on the prevalence of parasitic infections in children from Slovakia districts. The results of such studies are important and necessary, and in my opinion this paper should be published. However, the manuscript requires corrections, especially discussion section, in which there are mainly repetitions of the results, and similarities or differences between the other studies. So in fact there is no discussion on the main problem: what is the cause of these differences. Between communities observed in this study, between countries etc. Only one sentence at the end of the discussion section: "Our work is proof that poverty and poor hygiene contribute most to public health 456 problems." This is not enough. Lack of citations showing there is worse economic situation of e.g. Roma minority, lower education etc.

Other specific comments:

Line 97: Was it macro- or microscopic examination? What was the volume or mass of the samples?

Lines 99-100: Please describe shortly these methods.

Lines 121-128: Lack of sequence of primers for this reaction and lack of reference. In cited above reference No. 12 there are only primers for TPI gene fragment and SSU rRNA gene fragment.

Line 129: This sentence has no sense. It should be that water was negative control, and Giardia was positive control.

Line 163: The childrens' age is 1 month to 17 years in materials and methods section.

Line 166: How did the authors recognize that the eggs of Hymenolepis belonged to H. nana not H. diminuta? Of course there is much more higher probablity that the eggs were from H. nana, however this should be discussed in the discussion section.

Line 169: "0 to 2 years" or from 1 month to 2 years? Or to 24 months?

Line 238: This sentence is redundant in the results section.

Lines 381-387: These are repeated results. No discussion

Line 391: "In our study, parasitic infections are a serious problem" - what does it mean?

Line 394: "parasitic infectious infections" - what does it mean?

Lines 459-461: I think that the most important is education of the society.

Author Response

Response to the reviewer’s critique on the manuscript entitled: "Neglected Diseases - Parasitic Infections among Slovakian Children from Different Populations and Genotypes of Giardia duodenalis".

The authors are thankful for the Reviewer´s valuable suggestions. The manuscript has been completely revised as suggested by the Reviewer. Each particular concern is addressed point by point.

REVIEWER 1

Response to the reviewer’s critique on the manuscript entitled: "Neglected Diseases - Parasitic Infections among Slovakian Children from Different Populations and Genotypes of Giardia duodenalis".

The authors are thankful for the Reviewer´s valuable suggestions. The manuscript has been completely revised as suggested by the Reviewer. Each particular concern is addressed point by point.

Concern: The manuscript presents results of the epidemiological studies on the prevalence of parasitic infections in children from Slovakia districts. The results of such studies are important and necessary, and in my opinion this paper should be published. However, the manuscript requires corrections, especially discussion section, in which there are mainly repetitions of the results, and similarities or differences between the other studies. So in fact there is no discussion on the main problem: what is the cause of these differences. Between communities observed in this study, between countries etc. Only one sentence at the end of the discussion section: "Our work is proof that poverty and poor hygiene contribute most to public health 456 problems." This is not enough. Lack of citations showing there is worse economic situation of e.g. Roma minority, lower education etc.

Response: The text has been revised to address reviewer concern.

Other specific comments:

Concern: Line 97: Was it macro- or microscopic examination? What was the volume or mass of the samples?

Response: The information was added in the text.

Concern: Lines 99-100: Please describe shortly these methods.

Response: The paragraph describing the methods was added in the text.

Concern: Lines 121-128: Lack of sequence of primers for this reaction and lack of reference. In cited above reference No. 12 there are only primers for TPI gene fragment and SSU rRNA gene fragment.

Response: Thanks for your notes, text was corrected and revised. Text was corrected.

Concern: Line 129: This sentence has no sense. It should be that water was negative control, and Giardia was positive control.

Response: Text was corrected and revised.

Concern: Line 163: The childrens' age is 1 month to 17 years in materials and methods section.

Response: Text was corrected.

Concern: Line 166: How did the authors recognize that the eggs of Hymenolepis belonged to H. nana not H. diminuta? Of course there is much more higher probablity that the eggs were from H. nana, however this should be discussed in the discussion section.

Response: Thanks for your notes. We assume that these were H. nana eggs, because H. diminuta infection would require the consumption of cysticercoides from insects. While H. nana tapeworm infection can be caused without an intermediate host, via autoinfection.

Concern: Line 169: "0 to 2 years" or from 1 month to 2 years? Or to 24 months?

Response: Text was corrected and revised.

Concern: Line 238: This sentence is redundant in the results section.

Response:The sentence was omitted from the results section.

Concern: Lines 381-387: These are repeated results. No discussion.

Response: Text was corrected and revised.

Concern: Line 391: "In our study, parasitic infections are a serious problem" - what does it mean?

Response: Text was corrected and revised.

Concern: Line 394: "parasitic infectious infections" - what does it mean?

Response: Thanks for the comment. Text was corrected

Concern: Lines 459-461: I think that the most important is education of the society.

Response: We agree with reviewer. Text was corrected.

Reviewer 2 Report

Dear Authors,

The concept of your manuscript “Neglected Diseases - Parasitic Infections among Slovakian Children from Different Populations and Genotypes of Giardia duodenalis” is interesting and can provide information about the impact of different environmental and cultural conditions on the level of parasitism in children. I have no objection to the methods used. However, the overall presentation of the manuscript, in terms of result presentation, discussion and English language does not do any justice to the effort.

Major deficiencies

  1. Please, get correctly informed of what the terms frequency, prevalence, incidence and intensity of infection mean and how to use them properly. These terms have been used indiscriminately in the manuscript giving false impressions and incorrect descriptions.

  1. Terminology overall is problematic in the whole manuscript. Please do not use the word “germs” for parasites. What could “baby eggs” be (line 196) and what embryos of A. lumbricoides and E. vermicularis? I suppose eggs? Similarly, lines 2012-217. What is the term “marginalized group”? It appears for the first time here in Results. There is no description of what this is before. Marginalized groups, Marginalized settlements, Marginalized children, A lot of terms appearing suddenly in the results, referring to materials (actually to a part of the samples). Also, line 352: “125 models were positive” what does the term “model” stand for here?
  1. Please take good care of your writing. In many parts, the writing does not make any logical sense. Only a few examples:

-Line 46. The WHO identifies parasitic infections in humans as diseases that cause these infections to occur most frequently and in high prevalence in environments with increased efficacy in animal excrement.

-Line 55: The most common parasitic species in human are Ascaris lumbricoides, Trichuris trichiura, Ancylostoma duodenale, and also Necator americanus [5-7]. The most common intestinal parasite in humans and animals is a protozoan Giardia duodenalis.

-Line 72. Although these infections are generally underdiagnosed and considered less necessary for public health, they increasingly demonstrate the importance of significant morbidity or even mortality worldwide [11].

And there are a lot more.

  1. How exactly were the samples collected? Were the children brought to a hospital for general examination? Was it a door-to-door collection? Were all children clinically healthy? The Authors have to provide more data on the samples.

  1. Was the nested PCR for ß giardine gene performed on the basis of a previous publication? No reference is mentioned for this method.

  1. On what criteria were the age groups formed? Also, in which group the children of 2.5 or 5.5 etc., were included?

  1. In the results, there is a random passing from overall results to specific group results (and this without consistency, e.g. Line 184, there is no description of the findings in the minority group as it was the case above, for the majority group, i.e. what parasites were found and in what frequencies) and back to overall results. This should be corrected.

  1. Lines 212-222 The results are not clearly presented. Also, they are not consistent with Table 2 where they refer. E.g. X2 = 16.82; p <.0001 is not shown in Table 2.

  1. In the discussion, it is not clear at all on what basis was the literature selected. Some of the cited papers refer to adults (e.g. references 7 and 25) and other to children. Surveys that were not conducted in minority population children (e.g. reference 35) are compared to the present result in Roma children. In general, there is no consistency of the references used and mentioned in the discussion nor any obvious analogy with the present study (some of the cited papers refer to children, some others to the general population (not children), but all these are used in a random way, seemingly selected on the basis of infection percentages similar to the ones found in the present study, or other, unclear criteria.

Also, there is confusion in the Discussion, because the overall prevalences and findings are discussed and then separately the results from the Roma children, but not those from the non-Roma children.

Overall, the Discussion needs thorough revision and the presentation of other, similar to the present, studies must be made in a more comprehensive way.

  1. Line 411. Results cannot be introduced in the Discussion for the first time. So, if the authors decide to present the results of the two population groups separately, this must be done in the Results, in the Text and in a Table.

  1. The English language needs thorough revision.

Minor corrections

-Table 1. Total not Summary

-Pollution is not contamination. Please correct

-Line 53 change to protozoa and metazoa

-Line 85. The median value of ages is also important to provide here.

-Line 97. I believe you mean macroscopically.

-Line 212. Does this sentence say anything different than the previous? Also, you have to clarify at the beginning that the nested PCR was performed as previously described [12].

-Line 135. The sequencing of DNA and data analysis products from the PCR methods were purified ?? Rephrase, please.

-Line 164. Please also write the number of positive samples (125).

-Line 190. Refer to Table 2.

-Line 246. explanation of the abbreviations must be given the first time the terms appear in the text, not here

-Line 363. Is it the most common of all causes? Please provide the reference.

Author Response

Response to the reviewer’s critique on the manuscript entitled: "Neglected Diseases - Parasitic Infections among Slovakian Children from Different Populations and Genotypes of Giardia duodenalis".

The authors are thankful for the Reviewer´s valuable suggestions. The manuscript has been completely revised as suggested by the Reviewer. Each particular concern is addressed point by point.

REVIEWER 2

Response to the reviewer’s critique on the manuscript entitled: "Neglected Diseases - Parasitic Infections among Slovakian Children from Different Populations and Genotypes of Giardia duodenalis".

The authors are thankful for the Reviewer´s valuable suggestions. The manuscript has been completely revised as suggested by the Reviewer. Each particular concern is addressed point by point.

Dear Authors,

The concept of your manuscript “Neglected Diseases - Parasitic Infections among Slovakian Children from Different Populations and Genotypes of Giardia duodenalis” is interesting and can provide information about the impact of different environmental and cultural conditions on the level of parasitism in children. I have no objection to the methods used. However, the overall presentation of the manuscript, in terms of result presentation, discussion and English language does not do any justice to the effort.

 Major deficiencies

Concern: 1. Please, get correctly informed of what the terms frequency, prevalence, incidence and intensity of infection mean and how to use them properly. These terms have been used indiscriminately in the manuscript giving false impressions and incorrect descriptions.

Response: Thanks for the comment. The text was corrected as suggested.

Concern: 2. Terminology overall is problematic in the whole manuscript. Please do not use the word “germs” for parasites. What could “baby eggs” be (line 196) and what embryos of A. lumbricoides and E. vermicularis? I suppose eggs? Similarly, lines 2012-217. What is the term “marginalized group”? It appears for the first time here in Results. There is no description of what this is before. Marginalized groups, Marginalized settlements, Marginalized children, A lot of terms appearing suddenly in the results, referring to materials (actually to a part of the samples). Also, line 352: “125 models were positive” what does the term “model” stand for here?

Response: Text was corrected and revised. 

Concern: 3. Please take good care of your writing. In many parts, the writing does not make any logical sense. Only a few examples:

-Line 46. The WHO identifies parasitic infections in humans as diseases that cause these infections to occur most frequently and in high prevalence in environments with increased efficacy in animal excrement.

-Line 55: The most common parasitic species in human are Ascaris lumbricoides, Trichuris trichiura, Ancylostoma duodenale, and also Necator americanus [5-7]. The most common intestinal parasite in humans and animals is a protozoan Giardia duodenalis.

-Line 72. Although these infections are generally underdiagnosed and considered less necessary for public health, they increasingly demonstrate the importance of significant morbidity or even mortality worldwide [11].

And there are a lot more.

Response: Text was corrected and revised. 

Concern: 4. How exactly were the samples collected? Were the children brought to a hospital for general examination? Was it a door-to-door collection? Were all children clinically healthy? The Authors have to provide more data on the samples.

Response: The information was added in the text.

Concern: 5. Was the nested PCR for ß giardine gene performed on the basis of a previous publication? No reference is mentioned for this method.

Response: Thanks for your notes. A reference was added.

Concern: 6. On what criteria were the age groups formed? Also, in which group the children of 2.5 or 5.5 etc., were included?

Response: The paragraph, explaining it was added in the text. Aged categories were reformulated.

Concern: 7. In the results, there is a random passing from overall results to specific group results (and this without consistency, e.g. Line 184, there is no description of the findings in the minority group as it was the case above, for the majority group, i.e. what parasites were found and in what frequencies) and back to overall results. This should be corrected.

Response: Text was corrected as suggested.

Concern: 8. Lines 212-222 The results are not clearly presented. Also, they are not consistent with Table 2 where they refer. E.g. X2 = 16.82; p <.0001 is not shown in Table 2.

Response: The text was corrected and revised. 

Concern: 9. In the discussion, it is not clear at all on what basis was the literature selected. Some of the cited papers refer to adults (e.g. references 7 and 25) and other to children. Surveys that were not conducted in minority population children (e.g. reference 35) are compared to the present result in Roma children. In general, there is no consistency of the references used and mentioned in the discussion nor any obvious analogy with the present study (some of the cited papers refer to children, some others to the general population (not children), but all these are used in a random way, seemingly selected on the basis of infection percentages similar to the ones found in the present study, or other, unclear criteria.

Response: The text has been revised to address reviewer concern.

Concern: Also, there is confusion in the Discussion, because the overall prevalences and findings are discussed and then separately the results from the Roma children, but not those from the non-Roma children.

Response: The text was corrected and revised. 

Concern: Overall, the Discussion needs thorough revision and the presentation of other, similar to the present, studies must be made in a more comprehensive way.

Response: Discussion has been revised.

Concern: Line 411. Results cannot be introduced in the Discussion for the first time. So, if the authors decide to present the results of the two population groups separately, this must be done in the Results, in the Text and in a Table.

Response: Text was corrected as suggested.

Concern: The English language needs thorough revision.

Response: The English language was corrected..

 Minor corrections

Concern: Table 1. Total not Summary

Response: Corrected as suggested.

Concern: Pollution is not contamination. Please correct

Response: Corrected as suggested.

Concern: Line 53 change to protozoa and metazoa

Response: Corrected as suggested.

Concern: Line 85. The median value of ages is also important to provide here.

Response: Corrected as suggested.

Concern: Line 97. I believe you mean macroscopically

Response: Corrected as suggested.

Concern: Line 212. Does this sentence say anything different than the previous? Also, you have to clarify at the beginning that the nested PCR was performed as previously described [12].

Response: Corrected as suggested.

Concern: Line 135. The sequencing of DNA and data analysis products from the PCR methods were purified ?? Rephrase, please.

Response: Corrected as suggested.

Concern: Line 164. Please also write the number of positive samples (125).

Response: Corrected as suggested.

Concern: Line 190. Refer to Table 2.

Response: Corrected as suggested.

Concern: Line 246. explanation of the abbreviations must be given the first time the terms appear in the text, not here

Response: Corrected as suggested  

Concern: Line 363. Is it the most common of all causes? Please provide the reference. Response: Corrected as suggested

Reviewer 3 Report

This is an interesting article which focuses on parasite infections in a children population in a Central European country, namely Slovakia, and that reveals the genotypes of Giardia duodenalis present in the population analysed.

Nevertheless, the Ms requires a major revision to improve it and to make its eventual publication in Microorganisms possible.

The term “parasitic germs” is not adequate to define protozoan and helminth parasites.

Introduction

  • The 1st and the 2nd paragraphs should be moved to the end of this section, together with the last paragraph.
  • Paragraph #3 is written in telegram style with sentences that are not linked between them. It should be re-written.

Material and Methods

  • The 1st paragraph should be entitled as a subsection such as “Sample collection”.
  • Authors stated in the abstract that age ranges from 2 months to 17 years. However, in this section the youngest child analysed is 1 month old. This age range should be clarified.
  • Age classes should be explained in this first subsection.
  • A table including categories of origin in lines and age classes, sex, and match with population in columns will improve this subsection.
  • In the line 78 of the first paragraph of the subsection “Coprological diagnostic methods” it should say “(oo)cysts or cysts and oocysts but not oocysts only.
  • The two coprological methods used should briefly be explained and referenced.
  • The authors should mention as one of the limitations of the study that only a single sample of faeces was examined, and that, consequently, the prevalence of parasitation or even the presence of other parasite species could be expected. Moreover, another limitation that should be mentioned is that trofozoites were not preserved after the conservation of faeces at 4ºC between 24-48 hours.
  • Although the statistical analysis is correct, maybe if the authors use a Binary Logistic Regression, the resulting model could include more than one variable, but for that the number of age groups has to be reduced.

Results

  • The first two sentences of the Results section, as well as the first sentences of the 3rd and 4th paragraphs, belong to the Material and Methods section. Moreover, this is the first time that the authors mention the total number of faecal samples analysed.
  • Lines 206-208. The correct sentence should be “Differences between prevalences are statistically significant ...” because the minority population is not significantly more infected.
  • Lines 212-213. The correct sentence should be “Differences between prevalences were statistically significant...” but not the difference was significantly high.
  • In “P” results write 0.0001 not .0001, it can be confusing.
  • When age groups are analysed and their prevalences compared, the degree of freedom (df) of the χ2 test is = 4. Thus, the χ2 value needed to obtain a significant P value is too high. Maybe the number of age classes can be reduced. Moreover, prevalence between the first 4 groups are very similar, but I am sure that if the oldest group is compared separately with each other age group, in all cases the P value will be significant and the OR could be stated. This result will be very important to postulate the evolution of the parasitic prevalence between age groups and to elucidate/confirm the habits, mainly hygienic habits, to favour the infection route.
  • In subsection 3.1 “Molecular diagnosis of Giardia duodenalis in children”, the first sentence is not a result. Moreover, this subsection needs to be more clearly written and be more specific. It is difficult to follow, and readers not familiar with molecular diagnosis could find it difficult to understand these results.

Discussion

  • This section is mostly a comparison with other results obtained both in other European countries and in some developing countries. However, there is no real discussion of their own results, without explanations or proposals which explain them, and the routes of infection related to poor hygienic conditions, etc. Moreover, the majority of results are repeated once again. This section should be re-written.

Author Response

Response to the reviewer’s critique on the manuscript entitled: "Neglected Diseases - Parasitic Infections among Slovakian Children from Different Populations and Genotypes of Giardia duodenalis".

The authors are thankful for the Reviewer´s valuable suggestions. The manuscript has been completely revised as suggested by the Reviewer. Each particular concern is addressed point by point.

REVIEWER 3

Response to the reviewer’s critique on the manuscript entitled: "Neglected Diseases - Parasitic Infections among Slovakian Children from Different Populations and Genotypes of Giardia duodenalis".

The authors are thankful for the Reviewer´s valuable suggestions. The manuscript has been completely revised as suggested by the Reviewer. Each particular concern is addressed point by point.

This is an interesting article which focuses on parasite infections in a children population in a Central European country, namely Slovakia, and that reveals the genotypes of Giardia duodenalis present in the population analysed.

Nevertheless, the Ms requires a major revision to improve it and to make its eventual publication in Microorganisms possible.

Concern: The term “parasitic germs” is not adequate to define protozoan and helminth parasites.

Response: The text was corrected.

Introduction

Concern: The 1st and the 2nd paragraphs should be moved to the end of this section, together with the last paragraph.

Response: The paragraphs were omitted from the manuscript.

Concern: Paragraph #3 is written in telegram style with sentences that are not linked between them. It should be re-written.

Response: The text was corrected.

Material and Methods

Concern: The 1st paragraph should be entitled as a subsection such as “Sample collection”.

Response: The text was corrected.

Concern: Authors stated in the abstract that age ranges from 2 months to 17 years. However, in this section the youngest child analysed is 1 month old. This age range should be clarified. Response: Text was corrected and revised as suggested.

Concern: Age classes should be explained in this first subsection.

Response: The text was corrected.

Concern: A table including categories of origin in lines and age classes, sex, and match with population in columns will improve this subsection.

Response: The table was corrected.

Concern: In the line 78 of the first paragraph of the subsection “Coprological diagnostic methods” it should say “(oo)cysts or cysts and oocysts but not oocysts only.

Response: The text was corrected.

Concern: The two coprological methods used should briefly be explained and referenced.

Response: Text was corrected as suggested.

Concern: The authors should mention as one of the limitations of the study that only a single sample of faeces was examined, and that, consequently, the prevalence of parasitation or even the presence of other parasite species could be expected. Moreover, another limitation that should be mentioned is that trofozoites were not preserved after the conservation of faeces at 4ºC between 24-48 hours.

Response: Text was corrected as suggested.

Concern: Although the statistical analysis is correct, maybe if the authors use a Binary Logistic Regression, the resulting model could include more than one variable, but for that the number of age groups has to be reduced.

Response: Text was corrected as suggested.

Results

Concern: The first two sentences of the Results section, as well as the first sentences of the 3rd and 4th paragraphs, belong to the Material and Methods section. Moreover, this is the first time that the authors mention the total number of faecal samples analysed. 

Response: Text was corrected as suggested.

Concern: Lines 206-208. The correct sentence should be “Differences between prevalences are statistically significant ...” because the minority population is not significantly more infected.

Response: Text was corrected as suggested.

Concern: Lines 212-213. The correct sentence should be “Differences between prevalences were statistically significant...” but not the difference was significantly high.

Response: Text was corrected as suggested.

Concern: In “P” results write 0.0001 not .0001, it can be confusing

Response: Corrected as suggested

Concern: When age groups are analysed and their prevalences compared, the degree of freedom (df) of the χ2 test is = 4. Thus, the χ2 value needed to obtain a significant P value is too high. Maybe the number of age classes can be reduced. Moreover, prevalence between the first 4 groups are very similar, but I am sure that if the oldest group is compared separately with each other age group, in all cases the P value will be significant and the OR could be stated. This result will be very important to postulate the evolution of the parasitic prevalence between age groups and to elucidate/confirm the habits, mainly hygienic habits, to favour the infection route.

Response: Text was corrected as suggested.

Concern: In subsection 3.1 “Molecular diagnosis of Giardia duodenalis in children”, the first sentence is not a result. Moreover, this subsection needs to be more clearly written and be more specific. It is difficult to follow, and readers not familiar with molecular diagnosis could find it difficult to understand these results.

Response: We have reformulated some sentences in the subsection of the molecular part to remove ambiguity and formulate the results as more understandable. We have deleted the first sentence of subsection as recommended. In addition, to derive some output regarding the geographical distribution of G. duodenalis variants, we have added the following sentence to the final part devoted to the bg gene in subsection 3.1:

“No geographically induced preference towards any of the assemblages A, B was detected when comparing their distribution in two districts (Košice, Rožňava) with several analyzed samples in tpi.”

We have deleted the first sentence of subsection as recommended.

Discussion

Concern: This section is mostly a comparison with other results obtained both in other European countries and in some developing countries. However, there is no real discussion of their own results, without explanations or proposals which explain them, and the routes of infection related to poor hygienic conditions, etc. Moreover, the majority of results are repeated once again. This section should be re-written.

Response: Discussion was re-written. 

Round 2

Reviewer 2 Report

The authors have considerably improved the manuscript which I now believe is suitable for publication, after minor corrections. Please find below the specific comments.

-Line 162. I believe 2,500 to be rounds per minute. Please correct by giving the right number of g.

-Table 1. Regarding the total number of positive samples, please add an asterisk after 125 and add a footnote explaining that this number includes 17 cases of mixed infection.

-Paragraph at line 467: Please also refer to the different morphology of the eggs of these species. Unlike that of H. nana, the oncosphere of H. diminuta eggs does not bear conspicuous knobs and filaments at the poles. (Possible reference: https://www.sciencedirect.com/book/9780124159150/human-parasitology). These eggs cannot be confused between them.

Author Response

Reviewer 2

The authors are thankful again for the Reviewer´s valuable suggestions. The manuscript has been revised as suggested by the Reviewer. Each particular concern is addressed point by point.

Comments and Suggestions for Authors

The authors have considerably improved the manuscript which I now believe is suitable for publication, after minor corrections. Please find below the specific comments.

Concern: Line 162. I believe 2,500 to be rounds per minute. Please correct by giving the right number of g.

Response: Thanks for your notes, text was corrected.

Concern: Table 1. Regarding the total number of positive samples, please add an asterisk after 125 and add a footnote explaining that this number includes 17 cases of mixed infection.

Response: A footnote explaining it was added.

Concern: Paragraph at line 467: Please also refer to the different morphology of the eggs of these species. Unlike that of H. nana, the oncosphere of H. diminuta eggs does not bear conspicuous knobs and filaments at the poles. (Possible reference: https://www.sciencedirect.com/book/9780124159150/human-parasitology). These eggs cannot be confused between them.

Response: The information was added in the text.

Reviewer 3 Report

The authors followed most of my suggestions and the Ms has been improved substantially. although they did not include in the Material and Methods section the table that I proposed and they did not include the limitations of the study (see my previouys comments). Therefore, the decision on whether to include these two points is up to the ediorial team.

Moreover, there are some typographical mistakes that should be corrected.

Author Response

Reviewer 3

The authors are thankful again for the Reviewer´s valuable suggestions. The manuscript has been revised as suggested by the Reviewer. Each particular concern is addressed point by point.

Concern: The authors followed most of my suggestions and the Ms has been improved substantially. although they did not include in the Material and Methods section the table that I proposed and they did not include the limitations of the study (see my previouys comments). Therefore, the decision on whether to include these two points is up to the ediorial team.

Response: Table was added in the text.

Concern: Previous comment from the first round of the review: The authors should mention as one of the limitations of the study that only a single sample of faeces was examined, and that, consequently, the prevalence of parasitation or even the presence of other parasite species could be expected. Moreover, another limitation that should be mentioned is that trofozoites were not preserved after the conservation of faeces at 4ºC between 24-48 hours.

Response: Thanks for the comment. In our study we examined only the faeces, not the contents of the intestines. Trophosoites are in the intestine and again encystize to cysts at the interface of the jejunum and ileum. Cysts are then excreted in the faeces. Cysts were preserved after the conservation of faeces at 4ºC between 24-48 hours.

Concern: Moreover, there are some typographical mistakes that should be corrected.

Response: Mistakes were corrected.